# Evaluation of the Activity of Amino Chalcone Against *Staphylococcus* Strains Harboring Efflux Pumps

**DOI:** 10.3390/ph18111629

**Published:** 2025-10-28

**Authors:** Isydório Alves Donato, Cristina Rodrigues dos Santos Barbosa, Antonio Henrique Bezerra, Suieny Rodrigues Bezerra, Ray Silva Almeida, Cícera Datiane de Morais Oliveira-Tintino, Isaac Moura Araújo, Ewerton Yago de Sousa Rodrigues, Maria Yasmin Cândido de Oliveira, Francisco Ferdinando Cajazeiras, Jayza Maria Lima Dias, Jesyka Macedo Guedes, Jéssica Híade Silva Cristino, Emmanuel Silva Marinho, Márcia Machado Marinho, Hélcio Silva dos Santos, Henrique Douglas Melo Coutinho, Saulo Relison Tintino, Irwin Rose Alencar de Menezes, Francisco Assis Bezerra da Cunha

**Affiliations:** 1Laboratory of Semiarid Bioprospecting (LABSEMA), Regional University of Cariri (URCA), Crato 63105-000, CE, Brazil; isydorio.donato@urca.br (I.A.D.); cristinase75@gmail.com (C.R.d.S.B.); henriquebezerra.urca@gmail.com (A.H.B.); suieny.urca@gmail.com (S.R.B.); ewerton.sousarodrigues@urca.br (E.Y.d.S.R.); yasmim.oliveira@urca.br (M.Y.C.d.O.); saulo.tintino@urca.br (S.R.T.); 2Laboratory of Microbiology and Molecular Biology, Department of Chemical Biology, Regional University of Cariri (URCA), Crato 63105-000, CE, Brazil; ray.almeida@urca.br (R.S.A.); datianemorais@hotmail.com (C.D.d.M.O.-T.); isaac.moura@urca.br (I.M.A.); hdmcoutinho@gmail.com (H.D.M.C.); 3Chemistry Program, Vale do Acaraú State University (UVA), Sobral 62040-370, CE, Brazil; ferdinandocajazeiras@gmail.com (F.F.C.); jaizadias48@gmail.com (J.M.L.D.); jesyka.mg@gmail.com (J.M.G.); jessica.hiade@aluno.uece.br (J.H.S.C.); helciodossantos@gmail.com (H.S.d.S.); 4Graduate Program in Natural Sciences, State University of Ceará (UECE), Fortaleza 60714-903, CE, Brazil; emmanuel.marinho@uece.br (E.S.M.); marinho.marcia@gmail.com (M.M.M.); 5Brazil National Institute of Science and Technology of the Health Economic–Industrial Complex (iCEIS), Laboratory of Phamacology and Chemical Molecular, Department of Chemical Biology, Regional University of Cariri (URCA), Crato 63105-000, CE, Brazil

**Keywords:** *Staphylococcus aureus*, chalcone, bacterial resistance, efflux pump, docking, ADMET

## Abstract

**Background/Objectives:** The increasing prevalence of multidrug-resistant *Staphylococcus aureus* represents a major clinical challenge, primarily driven by the acquisition of multiple resistance mechanisms. Among these, efflux pumps such as NorA play a pivotal role in quinolone resistance by promoting active drug extrusion and reducing intracellular antibiotic levels. This study evaluated the synthetic chalcone CMA4DMA as a potential NorA efflux pump inhibitor and modulator of bacterial resistance. **Methods:** Antimicrobial susceptibility assays were conducted against *S. aureus* SA1199 (wild-type) and SA1199B (NorA-overexpressing) strains. The minimum inhibitory concentration (MIC) of CMA4DMA and its modulatory effects on norfloxacin and ethidium bromide were determined. Efflux inhibition was assessed by ethidium bromide accumulation and SYTOX Green assays. Molecular docking and in silico ADMET analyses were performed to predict binding affinity and pharmacokinetic parameters. **Results:** CMA4DMA exhibited no intrinsic antibacterial activity (MIC ≥ 1024 µg/mL) but reduced the MIC of norfloxacin from 32 to 8 µg/mL and that of ethidium bromide from 32 to 8 µg/mL in SA1199. In SA1199B, reductions from 64 to 16 µg/mL and from 64 to 32 µg/mL were observed, respectively. Fluorescence increased by 15% without affecting membrane integrity. Docking revealed a binding affinity of −7.504 kcal/mol, stronger than norfloxacin (−7.242 kcal/mol), involving key residues Leu218, Ile309, Arg310, and Ile313. ADMET data indicated high intestinal absorption (88.76%) and permeability (Papp = 1.38 × 10^−5^ cm/s). **Conclusions:** CMA4DMA effectively restored norfloxacin susceptibility in resistant *S. aureus* strains, highlighting its potential as a promising scaffold for developing novel efflux pump inhibitors and antibiotic adjuvants.

## 1. Introduction

Antibiotic resistance constitutes one of the greatest contemporary challenges to public health, impacting infection-related mortality, prolonging the course of diseases, increasing treatment costs, and reducing social productivity. In addition, it compromises the effectiveness of medical interventions that rely on antimicrobials, such as prophylactic surgeries and chemotherapy [1,2,3,4]. The emergence of resistance largely results from the adaptive capacity of bacterial populations, intensified by the indiscriminate use of antibiotics, which favors the acquisition and dissemination of resistance mechanisms.

Among multidrug-resistant pathogens, *Staphylococcus aureus* (*S. aureus*) stands out due to its high virulence, ability to evade immune responses, and increasing resistance to antimicrobials [5,6,7,8]. This versatile microorganism can cause anything from mild skin lesions to severe diseases such as sepsis, pneumonia, and toxic shock syndrome through the production of multiple virulence factors, including toxins, adhesion proteins, and immune evasion systems such as protein A, superantigenic toxins, and the agr quorum-sensing system [9].

Resistance in *S. aureus* arises both from spontaneous genetic mutations and from the acquisition of resistance genes via horizontal transfer. Mobile genetic elements such as plasmids, transposons, pathogenic islands (SaPIs), and staphylococcal chromosomal cassettes (SCCs) enable the rapid dissemination of resistance determinants [10]. Among the most important mechanisms are enzymatic inactivation of antibiotics, modification of molecular targets, reduced membrane permeability, and activation of efflux pumps [11,12,13]. Processes such as conjugation, natural transformation, and phage-mediated transduction contribute to the emergence of highly virulent and resistant strains.

Among these mechanisms, efflux pumps stand out as central resistance systems, promoting the active extrusion of antibiotics and other toxic compounds [14]. Studies indicate that the *S. aureus* genome encodes at least 31 multidrug efflux pumps distributed among the major transporter families [15]. The Major Facilitator Superfamily (MFS) has received particular attention, especially the NorA pump, responsible for the extrusion of fluoroquinolones and other drugs. Its function operates through a proton/drug antiport mechanism, which may be constitutively expressed or induced by exposure to antimicrobials [16,17,18].

Given the scarcity of new antimicrobials, alternative strategies have been explored, including the use of efflux pump inhibitors (EPIs) [19]. These compounds block drug extrusion, potentially reversing efflux-mediated resistance, reducing required doses, minimizing adverse effects, and restoring the efficacy of previously ineffective antibiotics [19,20,21,22]. EPIs can be natural or synthetic, and compounds derived from medicinal plants, such as chalcones, have shown promise due to their structural diversity and broad spectrum of biological activities [21,23].

The literature reports various efflux pump inhibitors (EPIs), both synthetic and natural, capable of restoring antibiotic effectiveness against resistant strains. Among the synthetic inhibitors, PAβN (phenylalanyl-arginyl-β-naphthylamide) is notable, functioning through competitive inhibition and gene regulation [24,25]; CCCP (carbonyl cyanide-m-chlorophenylhydrazone), which disrupts ATP synthesis [26]; NMP (1-(1-naphthylmethyl)-piperazine), which hinders the proper assembly of pumps [27]; and MBX2319, a pyrazolopyridine derivative with proven activity against *E. coli* [28]. In the natural inhibitors group, silybin stands out for reducing the expression of NorA and qacA/B genes in *S. aureus* [29]; boeravinone B, which enhances ciprofloxacin’s action [30]; curcumin, an inhibitor of the TetK pump [31]; berberine and columbamine, which interfere with ATP synthesis [32]; and reserpine, which binds directly to transport proteins [33]. Additionally, essential oils from Origanum vulgare and Nigella sativa have shown potential in modulating efflux gene expression and altering bacterial membrane permeability [34,35].

Although numerous natural and synthetic compounds have been described as efflux pump inhibitors, the search for effective, safe, and structurally versatile molecules still represents a significant challenge in combating bacterial resistance [22]. In this context, chalcones stand out for their broad spectrum of biological activities and their ability to modulate resistance mechanisms [36]. These molecules, widely distributed in flowers, leaves, fruits, and roots, can be obtained by chemical or semi-synthetic synthesis [37], exhibit low toxicity, a characteristic yellow coloration, and act as precursors of various bioactive compounds [38,39]. Studies indicate that chalcones display antioxidant, anti-inflammatory [40], antimicrobial [39], anticancer [38], and antiviral activities [41], in addition to enhancing antibiotic action and reducing virulence factors such as biofilm formation and toxin production [42,43,44].

However, investigations involving specific synthetic derivatives, such as the chalcone CMA4DMA, remain scarce. Thus, the present study aimed to evaluate the potential of the synthetic chalcone CMA4DMA as an inhibitor of the NorA efflux pump in *S. aureus* through an integrated in vitro and in silico approach, aiming to contribute to the development of new therapeutic strategies against resistant bacterial infections.

## 2. Results

### 2.1. In Vitro Antibacterial Activity

#### 2.1.1. Evaluation of Minimum Inhibitory Concentration (MIC)

Upon analyzing direct antibacterial activity, it was observed that the chalcone exhibited no direct activity in the in vitro minimum inhibitory concentration (MIC) assay, with values equal to or greater than 1024 µg/mL for all strains, which is considered irrelevant for advancing to clinical studies. Regarding the standard efflux pump inhibitor Carbonyl Cyanide M-Chlorophenyl Hydrazone (CCCP), MIC values of 8 µg/mL for strain 1199 and 16 µg/mL for strain 1199B were identified.

#### 2.1.2. Efflux Pump Inhibition by Modulation of Antibiotic and Ethidium Bromide MIC

In the analysis of potential MIC changes in antibiotics and ethidium bromide in the wild-type 1199 strain, it was observed that the chalcone was able to reduce the MIC of both the antibiotic and ethidium bromide. The antibiotic MIC decreased from 32 µg/mL to 8 µg/mL, and when the activity of ethidium bromide was evaluated in combination with the adjuvant, a similar reduction from 32 µg/mL to 8 µg/mL was observed. The positive control also reduced the MIC from 32 µg/mL to 0.5 µg/mL in both associations, as shown in Figure 1 and Table 1.

Regarding the MIC changes in the NorA-overexpressing 1199B strain, it was observed that the chalcone, as an adjuvant, was able to reduce the MIC of both the antibiotic and ethidium bromide: from 64 µg/mL to 16 µg/mL for the antibiotic and from 64 µg/mL to 32 µg/mL for ethidium bromide. The positive control CCCP reduced the MIC from 64 µg/mL to 16 µg/mL for both the antibiotic and ethidium bromide, as shown in Figure 2 and Table 1.

#### 2.1.3. Evaluation of NorA Efflux Pump Inhibition by Ethidium Bromide Fluorescence Measurement

In the assessment of ethidium bromide (EtBr) accumulation in *S. aureus* 1199B, CMA4DMA at 200 μg/mL caused a significant 15% increase in EtBr fluorescence compared to the negative control. A similar significant result was observed with the standard efflux pump inhibitor CCCP, which increased fluorescence by 21% in the 1199B strain (Figure 3; Table 2).

#### 2.1.4. Assessment of Increased Bacterial Membrane Permeability

It was observed that CMA4DMA at 200 μg/mL and 100 μg/mL did not increase SYTOX Green fluorescence intensity compared to the negative control in the tested strains, showing no statistically significant difference from the negative control. This indicates that the compound does not affect membrane permeability. The positive control, polymyxin B at 200 μg/mL and 100 μg/mL, significantly increased fluorescence intensity by 39% and 29% in *S. aureus* 1199B. Fluorescence measurements of SYTOX Green showed no statistically significant changes (Figure 4; Table 3).

### 2.2. Molecular Docking Analysis

As a case in point, the ligand–protein interactions formed between the Fab antigenic sequence and the NorA efflux pump residues were plotted (see Figure 5A). It was observed that the aromatic portions of the antigen formed a series of hydrophobic interactions with alkyl or aromatic side chain residues (Figure 5). Among these, it is possible to highlight the residues of Phe140, Ile244, Phe303 (Figure 5B), which are characterised as interaction residues of known substrates of the NorA efflux pump, such as norfloxacin [45]. These are fundamental residues in the identification of new inhibitors of NorA from strain 1199B [46].

When analyzing the docking simulation poses relative to the NorA efflux pump of *S. aureus* 1199B, it was observed that the chalcone CMA4DMA bound within the domain where the FAB antigen is complexed, occupying the same binding cavity as the antibiotics Norfloxacin and Ciprofloxacin, designated as the NorA efflux pump binding domain (Figure 6A). The cavity of this domain has a predicted molecular surface area of 1260.43 Å^2^ (Figure 6B), where a greater spatial distance from the CCCP (carbonyl cyanide m-chlorophenyl hydrazone) control can be noted, corroborating the experimental analyses, in which CCCP reduced the inhibitory concentration of Norfloxacin and EtBr. At the end of the cycle of 20 independent docking poses, CMA4DMA complexed with the NorA efflux pump with a binding affinity (BA) of –7.504 kcal/mol, which was energetically more favorable than Norfloxacin, which exhibited a BA of –7.242 kcal/mol (Table 4). All selected best-poses performed within a favorable statistical threshold of RMSD <2.0 Å, ensuring reproducibility of the docking simulation protocol [47].

Analysis of ligand–receptor interactions revealed that CMA4DMA (green) formed hydrophobic interactions with the alkyl portions of residues Leu218, Ile309, Arg310, and Ile313, whereas Norfloxacin (blue) tended to form hydrophobic interactions with aromatic residues, including Phe140 and Phe303, which are neighboring residues within the binding domain cavity (Figure 6C) [48]. When compared with EtBr (Figure 6D), distinct interactions were observed, although near CMA4DMA, formed between the control and aromatic residues (Phe140, Tyr225, and Phe303). It is noteworthy that a hydrogen bond interaction was formed between the NH_2_ group of CMA4DMA and the polar portion of Thr211, where the calculated donor–acceptor distance of 2.28 Å indicates moderate-to-weak polar interactions (2.5 < d[Å] ≤ 3.1), with a stronger binding compared to Norfloxacin (d = 2.80 Å) [49].

### 2.3. In Silico ADMET Study

#### 2.3.1. Cell Permeability Prediction

In the predictive ADMET analysis, effective cellular permeability (Papp) in Caco-2 cells showed a strong correlation with the calculated physicochemical parameters of lipophilicity (logP) and aqueous solubility (logS). For the predicted parameters, a statistical correlation of r = 0.93 was calculated, indicating a directly proportional relationship between Caco-2 Papp and logP. The compound CMA4DMA occupied a physicochemical space statistically closer to the controls EtBr and CCCP, where logPapp indices increased from –5.2 when logP exceeded 2.0 (Figure 7A). This indicates compounds with low affinity for aqueous environments (logS ≤ –3.0) and high cellular permeability.

In the *drug-like space radar* (Figure 7B), the calculated logP of 2.74 positioned CMA4DMA within the ideal lipophilicity range according to Pfizer’s biopharmaceutical classification system (–2 < logP ≤ 5). Moreover, its molecular weight of 328.4 g/mol fell within the appropriate drug-like range (200–500 g/mol) (Table 5). The TPSA value of 46.33 Å^2^, largely influenced by the NH_2_ group, was within the optimal threshold for improving pharmacokinetic properties and reducing in vivo toxicity risk (40–90 Å^2^). The QED score of 0.525 further supports the pharmacokinetic viability of the compound.

The predicted Papp value in Caco-2 cells was 1.38 × 10^−5^ cm/s, indicating higher intestinal permeability compared with Norfloxacin and Ciprofloxacin, which showed values below 1.0 × 10^−5^ cm/s (Table 6). In MDCK cells, predicted permeability values above 1.5 × 10^−5^ cm/s suggested that CMA4DMA, similarly to EtBr and CCCP, may present high permeability in more selective cell lines. The predicted probability of interaction with the efflux transporter P-gp indicated a low likelihood of CMA4DMA being a substrate, a characteristic associated with compounds of low polarity (TPSA ≤ 90 Å^2^). Regarding distribution, the predicted Vdss of 0.37 L/kg suggested that the compound may distribute relatively evenly between blood plasma and biological tissues. Plasma protein binding predictions indicated PPB >90%, likely due to the compound’s low polarity and the hydrophobic nature of the molecule, favoring binding within the hydrophobic cavities of serum proteins.

#### 2.3.2. Brain and Intestinal Permeation and Cardiotoxicity

In the molecular lipophilicity potential (MLP) map, the structural contributions that favor human intestinal absorption (HIA) and blood–brain barrier (BBB) permeability were identified. Surface analysis indicated that the NH_2_ group of CMA4DMA is highly polar and exhibits a large solvent-accessible surface area (red spectrum), whereas the substituted aromatic rings concentrate most of the compound’s lipophilic surface (blue spectrum), resulting in a calculated logP of 2.74, consistent with optimal affinity for lipophilic environments (Figure 8A).

This property supports an estimated HIA of 88.76% (Figure 8B), as lipophilic compounds more readily permeate cell membranes and thus display enhanced absorption. The moderate-to-high lipophilicity (logP > 2.0–3.0) also favors permeability in selective cell lines, such as MDCK cells, commonly applied in BBB permeability assays. In this context, the predicted logBB of 0.87 correlated with the predicted MDCK permeability (Table 6), aligning with data observed for at least 87% of CNS-active compounds in the ADMET-LMC database (Figure 8C).

Additionally, the lipophilic nature of the substituted aromatic rings was associated with potential inhibition of hERG (human Ether-à-go-go-Related Gene) channels. In silico prediction suggested that the α,β-unsaturated aromatic system contributes negatively to K^+^-dependent hERG channel inhibition, yielding an estimated IC_50_ of 6.104 µM and a pIC_50_ of 5.21 ± 0.35 (Figure 8D). The data similarity test further predicted a potential inhibition pKi of 5.43 with 33% similarity to known hERG inhibitors (Figure 8E) and a predicted pIC_50_ of 5.69 with 67% similarity to compounds documented as hERG inhibitors in the ADMET-LMC database (Figure 8F).

#### 2.3.3. Site of Metabolism and Rat Acute Toxicity

Prediction of the metabolic site indicated that CMA4DMA contains an –N(CH_3_)_2_ group, which contributes negatively (red color spectrum) to first-pass metabolism. This site was identified as an N-dealkylation site mediated by major CYP450 isoforms, including CYP2D6, CYP2C9, and CYP3A4 (Figure 9A). Such biotransformation may generate unstable aldehydes and free radicals capable of interacting with macromolecules such as DNA, potentially leading to hepatotoxic effects [50]. The data similarity test confirmed that this is a structurally specific metabolic site within acceptable similarity thresholds (Figure 9B) [51]. Predicted first-pass kinetics suggested a hepatic clearance (CLHepa) of 76.50 µL/min/10^6^ cells, indicating that CMA4DMA could be more rapidly metabolized by hepatocytes than Norfloxacin and Ciprofloxacin, which exhibited CLHepa values below 15 µL/min/10^6^ cells, consistent with greater metabolic stability and oral bioavailability (Table 6) [52].

At the microsomal level, predicted clearance values (CLMicro) indicated that Norfloxacin and Ciprofloxacin are metabolically more stable (≤8.0 mL/min/kg) compared with CMA4DMA, which showed a predicted CLMicro of 10.58 mL/min/kg, reflecting relative metabolic instability and faster first-pass excretion of the chalcone (Table 6) [53]. Toxicity predictions in rats revealed an oral LD_50_ of 1924 mg/kg, slightly below the general safety threshold (LD_50_ > 2000 mg/kg), suggesting possible toxicity due to metabolic activation (Figure 9C) [54]. For parenteral routes, the predicted LD_50_ values were <500 mg/kg for intraperitoneal (IP) and intravenous (IV) administration, indicating greater systemic toxicity when first-pass metabolism is bypassed. For subcutaneous administration, the predicted LD_50_ of approximately 893.7 mg/kg suggested that this route could be feasible, particularly for topical or localized infection therapies (Figure 9C).

## 3. Discussion

The lack of a relevant MIC does not entirely disqualify a substance for potential application in other studies against microorganisms, particularly when considering its possible use as an antibiotic adjuvant. In the case of efflux pump inhibitors, the absence of direct activity may even enhance their applicability, as previously noted in the literature [55].

Restoring the activity of antibiotics or other antibacterial agents through adjuvants is one of the main strategies currently employed to control strains harboring multiple mechanisms of resistance to antibacterial drugs or other bactericidal agents [56]. This is particularly relevant because the discovery of new effective antibiotics is increasingly rare. Furthermore, even when a new antibiotic is discovered, the likelihood that its indiscriminate use will lead to bacterial resistance is high [57]. Various methods exist to assess the ability of substances to restore antibiotic activity and, consequently, to identify the bacterial resistance mechanisms involved. Reduction in the minimum inhibitory concentration is one of the first analyses performed to indicate the restoration of antibiotic activity, typically using sub-inhibitory concentrations of the test substances in combination with an antibiotic or another antibacterial agent [58].

In the present study, as shown in Figure 1, CMA4DMA at sub-inhibitory concentrations was able to restore antibiotic activity by reducing the MIC, and the same effect was observed in combination with ethidium bromide. In the case of ethidium bromide, the MIC reduction is indicative of efflux pump inhibition, which is reinforced by the action of the positive control, the efflux pump inhibitor CCCP. It is also noteworthy that the SA1199 strain is a NorA efflux pump carrier with basal expression, and the observed MIC reduction in this strain may indicate inhibition of basal NorA pump expression, as illustrated in Figure 1 [59].

The SA1199B strain is widely described in the literature for its high resistance to norfloxacin and other quinolones [16,60]. Resistance to this class of antibiotics in *S. aureus* can occur via efflux pumps or by modifications in topoisomerase, the primary target of quinolones [61]. MIC reduction in this strain, as observed in SA1199, may be associated with inhibition of previously described resistance mechanisms. The SA1199B strain overexpresses the NorA pump, and the decrease in MIC against certain antibiotics strongly suggests inhibition of this transporter, particularly when its expression is high in the bacterial cell membrane. To rule out interference from other mechanisms, it is common practice to evaluate MIC reduction in ethidium bromide, a characteristic substrate of NorA, alongside the effect of CCCP as a reference efflux pump inhibitor. Thus, the results presented in Figure 2 suggest that CMA4DMA has potential as a NorA pump inhibitor.

Studies with natural chalcones isolated from *Arrabidaea brachypoda* [62] and synthetic chalcones [63] have shown that, similarly to CMA4DMA, chalcones exhibit no direct antibacterial activity but significantly reduce the MIC of norfloxacin in SA1199B strains and of ethidium bromide, indicating NorA pump inhibition. Docking experiments confirmed ligand overlap with the norfloxacin binding site in NorA. These results are particularly similar to those observed with CMA4DMA, where the amino group on the phenolic ring contributed positively to MIC reduction in NorA-harboring strains [62,63].

In addition to the studies mentioned, the same chalcone used in the present work was investigated against *S. aureus* strains K4100, carrying QacC and β-lactamases [64], carrying MepA [65]. No intrinsic activity (MIC ≥ 1024 µg/mL) was observed in either study. In the K4100 strain, CMA4DMA did not inhibit β-lactamase but significantly reduced the MIC of EtBr from 64 to 32 µg/mL. In the previously mentioned study against strain K2068, significant MIC reductions for ciprofloxacin and ethidium bromide were observed. Docking studies confirmed that the chalcone can bind to the same site occupied by the standard chlorpromazine inhibitor, interacting with key protein residues, which supports its potential as a MepA inhibitor, as it may in [65].

These studies complement and validate the results obtained in the present work, demonstrating that CMA4DMA consistently reduces efflux-mediated resistance in different *S. aureus* genetic backgrounds (NorA, MepA, and QacC), consolidating it as a promising antibacterial adjuvant candidate.

According to Figure 3, the increase in fluorescence due to greater intracellular ethidium bromide accumulation corroborates the MIC reduction results, supporting efflux pump inhibition. CMA4DMA was able to promote increased intracellular EtBr, making it more available to intercalate with bacterial DNA as a consequence of efflux pump inhibition, which normally extrudes the compound from the cell [66].

Studies of intracellular bromide accumulation using CMA4DMA in *S. aureus* 1199B are nonexistent, but there are studies with other chalcones that show inhibition of the efflux pump with increased fluorescence due to ethidium bromide accumulation, as in the study by [67], which evaluated a library of 117 chalcones for inhibitory activity of the NorA pump mediated by EtBr efflux. Furthermore, the study by [68] evaluated the action of chalcone derivatives capable of increasing EtBr accumulation in *S. aureus* by inhibiting efflux pumps. Therefore, these two studies corroborate the present study by demonstrating the ability of chalcones to increase the amount of intracellular bromide, related to efflux protein inhibition.

Therefore, according to Figure 4, the chalcone CMA4DMA did not alter membrane permeability, and thus its effect cannot be attributed to indirect efflux pump inhibition via changes in membrane fluidity. Although indirect inhibition through membrane alteration is one mode of action for compounds targeting efflux pumps, direct interaction with the pump is more common [69,70]. Exceptions exist, such as in the study by [71], where menadione was shown to act both directly on the efflux pump and indirectly on the plasma membrane, the latter also potentially disabling the pump. It is likely that CMA4DMA acts at a binding site on the NorA protein, as suggested by molecular docking studies.

The docking results indicate that CMA4DMA can effectively bind to the same NorA efflux pump domain where fluoroquinolone antibiotics such as Norfloxacin and Ciprofloxacin interact. The more favorable binding affinity of CMA4DMA compared to Norfloxacin suggests a higher energetic stability of this interaction, reinforcing its potential as a modulator of efflux activity. The affinity energy of CMA4DMA is higher (i.e., has a more negative value) when compared to natural compounds that bind to NorA and have the potential to inhibit strain 1199B. These potential increases as values decrease from <–5.0 kcal/mol [45,72].

The hydrophobic interactions of CMA4DMA with aliphatic residues (Leu218, Ile309, Arg310, and Ile313), distinct from the predominantly aromatic interactions observed for Norfloxacin and EtBr, reveal that CMA4DMA occupies the cavity in a differentiated binding mode. This divergence may favor synergistic effects with antibiotics, as the compound could alter the conformational dynamics of the binding pocket. This phenomenon has also been observed in molecular docking studies associated with strain 1199B inhibition [45], where NorA ligand candidates can act synergistically with other known substrates, such as Norfloxacin.

Molecular recognition was applied in the analysis of molecular docking simulations, where it was possible to observe that the hydrophobic nature of chalcone would affect the van der Waals dispersive forces in hydrophobic interactions with the protein. The results demonstrate the significant contribution of the aromatic rings of the ligand in establishing hydrophobic interactions with the alkyl portion of the residues Leu218, Ile309, and Ile313, thereby establishing a privileged scaffold that facilitates the selection of lead compounds that inhibit the NorA efflux pump [48,73].

The hydrogen bonding observed between CMA4DMA and Thr211 further strengthens the binding profile, as the shorter donor–acceptor distance indicates more stable polar interactions compared to those of Norfloxacin. Importantly, the spatial distance of CMA4DMA relative to the CCCP control corroborates experimental findings, where CCCP potentiated the activity of Norfloxacin and EtBr, suggesting a mechanistic overlap in efflux pump inhibition. The residue has been found to be associated with the binding of known NorA substrates, such as levofloxacin, within the hydrophobic cavity of the efflux pump [74].

Taken together, these findings support the hypothesis that CMA4DMA has a potential inhibitory effect on the NorA efflux pump, which may be potentiated in the presence of antibiotics. This interaction pattern highlights CMA4DMA as a promising candidate for further studies aimed at efflux pump modulation and antibiotic resistance reversal.

The predictive ADMET results indicate that CMA4DMA possesses physicochemical properties compatible with high cellular permeability, an attribute strongly correlated with lipophilicity (logP). Its placement in the *drug-like space radar*—within the thresholds of logP, MW, and TPSA—reinforces the compound’s alignment with drug-like oral properties.

The predicted Papp value in Caco-2 cells, superior to that of the reference antibiotics Norfloxacin and Ciprofloxacin, suggests that CMA4DMA may exhibit higher intestinal absorption and, consequently, improved oral bioavailability. Additionally, the prediction of high permeability in MDCK cells and the low probability of being a P-gp substrate strengthen the perspective of efficient intracellular accumulation, favoring its antimicrobial potential.

The predicted Vdss, within the acceptable range, indicates that the compound may achieve a suitable balance between plasma and tissue distribution, minimizing the risk of excessive accumulation in specific compartments. The high level of plasma protein binding (>90%), although typical of low-polarity lipophilic molecules, may influence the free fraction available to exert pharmacological activity. Nonetheless, this feature may also contribute to plasma stability and prolonged circulation time. Overall, the ADMET analysis suggests that CMA4DMA combines favorable permeability, distribution, and pharmacokinetic viability, presenting advantages over comparative antibiotics in terms of absorption and cellular transport.

The MLP map and ADMET predictions suggest that the structural balance between polar and lipophilic regions in CMA4DMA is a key determinant of its pharmacokinetic profile. The predominance of lipophilic aromatic rings combined with a single polar NH_2_ donor provides the compound with an optimal logP value of 2.74, which is associated with both high intestinal absorption and efficient membrane permeation. The predicted HIA of nearly 89% strongly supports the potential of CMA4DMA for oral administration, as lipophilicity is a critical determinant of absorption efficiency.

The predicted logBB and MDCK permeability values indicate that CMA4DMA may cross the BBB, displaying distribution patterns consistent with a large proportion of CNS-active compounds. While this property could support antimicrobial efficacy in tissues with selective barriers, it also raises concerns regarding potential off-target CNS effects.

A key safety consideration is the compound’s predicted interaction with hERG channels. The α,β-unsaturated aromatic system contributes to moderate hERG inhibition potential, with IC_50_ and pIC_50_ values suggesting partial blockade of K^+^ channels. The data similarity to known hERG inhibitors (33–67%) highlights a moderate but significant risk for cardiotoxicity, particularly in scenarios of chronic exposure. These predictions emphasize the need for further experimental validation, as inhibition of hERG channels is strongly associated with arrhythmias and other cardiac disorders. Together, these findings indicate that CMA4DMA displays a favorable absorption and distribution profile, but its potential to interact with cardiac ion channels warrants caution and targeted safety evaluation.

The metabolic prediction highlights the –N(CH_3_)_2_ group of CMA4DMA as a major liability, acting as a specific site for N-dealkylation mediated by CYP450 isoforms. While this pathway contributes to clearance, it also poses a potential safety risk due to the possible generation of reactive intermediates such as aldehydes and free radicals, which could contribute to hepatotoxicity. This aligns with the observed moderate oral LD_50_, which falls just below the threshold of non-toxic compounds, suggesting a risk of toxicity linked to metabolic activation.

Compared with Norfloxacin and Ciprofloxacin, CMA4DMA appears to be less metabolically stable, as reflected by its higher predicted hepatic and microsomal clearance values. This reduced metabolic stability implies lower oral bioavailability, as the compound would be more extensively metabolized during first-pass hepatic processing.

The predicted LD_50_ values across different administration routes emphasize the role of first-pass metabolism in mitigating systemic toxicity. While oral administration may partially protect against systemic adverse effects, bypassing hepatic metabolism through intravenous or intraperitoneal administration results in greater predicted toxicity. Interestingly, the subcutaneous route showed an intermediate LD_50_, suggesting potential viability for localized or topical applications, where systemic exposure could be minimized.

Taken together, these predictions suggest that although CMA4DMA has promising pharmacological attributes, its metabolic liability and associated toxicity risks require careful consideration. Optimization of structural features to reduce CYP-mediated N-dealkylation, or the exploration of alternative administration routes, may be essential for improving its therapeutic potential.

## 4. Materials and Methods

### 4.1. Chemical Synthesis

The chalcone CMA4DMA was synthesized through a Claisen–Schmidt condensation reaction under basic conditions, using 3-aminoacetophenone and 4-(dimethylamino)benzaldehyde [64]. The synthetic pathway is illustrated in Figure 1, while the chemical structure of CMA4DMA is presented in Figure 10.

### 4.2. Molecular Docking of NorA and CMA4DMA

For the theoretical investigation of the antimicrobial effect against *S. aureus*, the protein model of the NorA efflux pump from the 1199B strain was obtained from the RCSB Protein Data Bank repository (https://www.rcsb.org/), deposited under PDB ID 7LO7. This structure is classified as a transporter protein expressed in *E. coli*, with its three-dimensional conformation resolved by electron microscopy at a 3.74 Å resolution. For protein preparation prior to molecular docking simulations, AutoDockTools™ version 1.5.7 (https://autodocksuite.scripps.edu/adt/) was used to add hydrogen atoms, compute Gasteiger charges, and adjust the grid-box parameters. The grid-box was configured to cover the entire conformational space of the protein, with dimensions x = 54, y = 58, and z = 62, centered at x = 137.578, y = 138.065, and z = 156.646. Subsequently, AutoDock Vina™ (https://vina.scripps.edu/) was set to perform 50 independent simulations of 20 poses each, for all ligands optimized using the Merck Molecular Force Field method (MMFF94) in Avogadro 2 (https://two.avogadro.cc/). The validation criteria for the optimal pose of molecular docking simulations include low Root Mean Square Deviation (RMSD ≤ 2.0 Å) and low affinity energy (Rate ≤ −6.0 kcal/mol), with the binding pose of the characterised antibiotics norfloxacin and ciprofloxacin serving as a control [75].

### 4.3. In Silico ADMET Study

For the analysis of absorption, distribution, metabolism, excretion, and toxicity (ADMET) properties, a predictive protocol based on the quantitative estimate of drug-likeness (QED) was applied, as shown in Equation (1):(1)$$QED=exp\left(\frac{1}{n}\sum_{i=1}^{n}\;\;lnd_{i}\right)$$
where n is the number of physicochemical properties (i) calculated, and d is the desirability function, which considers the thresholds defined as follows: molecular weight (MW) ≤ 500 g/mol, lipophilicity (logP) ≤ 5, hydrogen bond donors (HBD) ≤ 5, hydrogen bond acceptors (HBA) ≤ 5, topological polar surface area (TPSA) ≤ 140 Å^2^, rotatable bonds (nRot) ≤ 10, aromatic rings ≤ 3, and structural alerts ≤ 1 (n = 8). The summation yields a score ranging from 0.0 (poor drug-likeness) to 1.0 (optimal drug-likeness) [76]. The alignment of these descriptors was analyzed according to Lipinski’s rules [77], the Pfizer rule [78], the Golden Triangle rule [79], and the GlaxoSmithKline (GSK) rule.

For the prediction of pharmacokinetic descriptors, the following servers were used: ADMETlab 3.0 (https://admetlab3.scbdd.com/), ADMET-AI (https://admet.ai.greenstonebio.com/), ADMET–LMC (http://qsar.chem.msu.ru/admet/), Pred-hERG (https://predherg.labmol.com.br/), and SwissADME (http://www.swissadme.ch/). The properties analyzed included effective cellular permeability (Papp), human intestinal absorption (HIA), blood–brain barrier (BBB) permeability, hepatic clearance (CLHepa), volume of distribution (Vdss), and cardiotoxicity via human Ether-à-go-go-Related Gene (hERG) inhibition.

For the prediction of metabolic sites, the servers SMARTCyp (https://smartcyp.sund.ku.dk/mol_to_som), FAME 3 (https://nerdd.univie.ac.at/fame3), and StopTox (https://stoptox.mml.unc.edu/) were employed. The analysis of fragments susceptible to cytochrome P450 (CYP450)-dependent metabolism was correlated with the prediction of lethal dose (LD50) in rats using the GUSAR Online server (https://www.way2drug.com/gusar/).

### 4.4. In Vitro Antibacterial Activity

#### 4.4.1. Bacterial Strains

In this study, the *S. aureus* strains SA-1199 (wild-type) and SA-1199B were used. The SA-1199B strain overexpresses the NorA efflux protein, which is associated with resistance to fluoroquinolone-class substrates, such as norfloxacin.

The strains were kindly provided by Prof. S. Gibbons (University of London, Malet St, Bloomsbury, London WC1E 7HU, UK) and Prof. G. W. Kaatz (Wayne State University School of Medicine, Detroit, MI, USA) and maintained on blood agar (Difco Laboratories Ltd., São Paulo, Brazil). The strains are preserved by cryopreservation at −80 °C in glycerol solution at the Laboratory of Microbiology and Molecular Biology, Regional University of Cariri (URCA). For experimental preparation, the strains were cultured for 24 h at 37 °C in brain–heart infusion agar (BHI Agar, Acumedia Manufacturers Inc., Lansing, Michigan).

#### 4.4.2. Culture Media

The culture media used for the microbiological assays were as follows: Brain Heart Infusion Agar (BHI Agar, Acumedia Manufacturers Inc.), prepared according to the manufacturer’s instructions, and Brain Heart Infusion (BHI, Acumedia Manufacturers Inc.) prepared at a 10% concentration.

#### 4.4.3. Substances

The antibiotics (norfloxacin and ciprofloxacin) and the chalcone CMA4DMA were dissolved in dimethyl sulfoxide (DMSO) and sterile water, with the DMSO proportion kept below 5%. Ethidium bromide (EtBr) was dissolved in sterile distilled water, while carbonyl cyanide m-chlorophenyl hydrazone (CCCP) was dissolved in methanol/water (1:1, *v*/*v*). All substances were prepared at a standard concentration of 1024 µg/mL.

#### 4.4.4. Determination of Minimum Inhibitory Concentration (MIC)

The MIC of the chalcone CMA4DMA was determined according to the broth microdilution method described by [80], with modifications. The strains used in the assays were cultured 24 h prior to the experiments. After this period, the bacterial inoculum was suspended in saline solution corresponding to 0.5 McFarland standard, approximately 1.5 × 10^8^ CFU/mL. Subsequently, microtubes were filled with 900 µL of BHI and 100 µL of the inoculum, and the plates were loaded with 100 µL of the final solution. Microdilution was performed with 100 µL of CMA4DMA in serial dilutions up to the penultimate well of the plate (1:1), with the last well used as a growth control. Compound concentrations ranged from 512 µg/mL to 8.0 µg/mL. After 24 h of incubation, readings were performed by adding 20 µL of resazurin (7-hydroxy-10-oxidophenoxazin-10-ium-3-one). Resazurin is oxidized in the presence of acidic medium caused by bacterial growth, inducing a color change from blue to pink [81]. The MIC was defined as the lowest concentration at which no visible growth was observed [82]. All assays were performed in triplicate in a laminar flow hood.

#### 4.4.5. Evaluation of Efflux Pump Inhibition by Modification of the Antibiotic and Ethidium Bromide MIC

To investigate whether the chalcone CMA4DMA acts as a potential NorA efflux pump inhibitor, a comparative study was conducted between the effects of standard pump inhibitors, evaluating their ability to reduce the MIC of EtBr and antibiotics. The standard inhibitor CCCP was used to validate NorA pump expression in the tested strain. Efflux pump inhibition was tested using sub-inhibitory concentrations (MIC/8) of both the inhibitors and the chalcone. In the assays, 150 µL of each bacterial inoculum, suspended in saline solution corresponding to 0.5 McFarland standard (approximately 1.5 × 10^8^ CFU/mL), was added together with the inhibitors and the chalcone (MIC/8) and completed with BHI. Subsequently, the mixtures were transferred to 96-well microdilution plates, to which 100 µL of antibiotic or EtBr in serial dilutions (1:1) ranging from 512 to 0.5 µg/mL were added. Plates were incubated at 37 °C for 24 h, and bacterial growth was assessed using resazurin (7-hydroxy-10-oxidophenoxazin-10-ium-3-one). Resazurin is oxidized in the presence of acidic medium resulting from bacterial growth, inducing a color change from blue to pink [81]. The MIC was defined as the lowest concentration at which no visible growth was observed [82]. MICs of the controls were determined using plates containing only antibiotics or EtBr. All tests were performed in triplicate in a laminar flow hood.

#### 4.4.6. NorA Efflux Pump Inhibitory Activity Assessed by Increased EtBr Fluorescence Emission

For this assay, the *S. aureus* 1199B strain expressing the NorA efflux pump was used. Strains were streaked on Mueller Hinton agar 24 h prior to the experiment and maintained in a bacteriological incubator at 37 °C. The inoculum was prepared in phosphate-buffered saline (PBS) to a 0.5 McFarland standard and distributed into black 96-well plates. In the test groups, CMA4DMA was added at final concentrations of 200 and 100 µg/mL. The negative control consisted of PBS prepared under the same dilution conditions as the test substance, followed by the addition of bacteria. The positive control consisted of CCCP at 200 µg/mL and 100 µg/mL. Plates were incubated for 1 h, after which EtBr (2 µg/mL) was added to all wells. The plates were then incubated again and fluorescence was measured using a Cytation 1 microplate reader (BioTek^®^, Winooski, VT, USA) with Gen5™ 3.11 software. Excitation was set at 530 nm and emission at 590 nm [80].

#### 4.4.7. Assessment of Bacterial Membrane Permeability Using SYTOX Green Fluorescence Assay

For this assay, the DNA-intercalating dye SYTOX Green was used. The bacterial inoculum of *S. aureus* 1199B was prepared and distributed into black 96-well plates. CMA4DMA was added at final concentrations of 200 and 100 µg/mL. Polymyxin B was used as positive control at 200 and 100 µg/mL, while phosphate-buffered saline (PBS) served as the negative control. Plates were incubated for 1 h. Subsequently, 100 µL of SYTOX Green at a final concentration of 1 µM was added. Plates were incubated for an additional hour, and fluorescence was measured using a Cytation 1 microplate reader (BioTek^®^, Winooski, VT, USA) with Gen5™ 3.11 software. Excitation and emission wavelengths were set at 485 nm and 528 nm, respectively. All assays were performed in triplicate [83].

#### 4.4.8. Statistical Analysis

The results were analyzed using one-way ANOVA, followed by Tukey’s post hoc test. Microbiological results for MIC reduction were expressed as geometric mean ± SD, while results from efflux pump inhibition mechanism assays were expressed as arithmetic mean ± SD. Values of *p* < 0.05 were considered statistically significant. GraphPad Prism 5.0 software was used for all analyses. All assays were performed in triplicate.

## 5. Conclusions

This research demonstrated, in vitro assays, that the chalcone CMA4DMA, although devoid of direct antibacterial activity (MIC ≥ 1024 µg/mL), significantly enhanced the action of norfloxacin and ethidium bromide in *S. aureus* strains carrying the NorA efflux pump. This potentiation was evidenced by reduced MIC values and increased intracellular EtBr fluorescence, without compromising bacterial membrane integrity. These experimental findings corroborate the *in silico* analyses, which revealed a strong interaction of CMA4DMA with the NorA binding site, supporting its role as a direct efflux pump inhibitor. Supported by the fact that the compound CMA4DMA showed a more stable binding to the NorA efflux pump of *S. aureus* 1199B (BA = −7.504 kcal/mol) than Norfloxacin, occupying the same binding site. Hydrophobic interactions and a hydrogen bond with Thr211 indicate higher affinity and inhibitory potential toward NorA. Additionally, CMA4DMA demonstrated optimal lipophilicity (logP = 2.74) and permeability (Papp = 1.38 × 10^−5^ cm/s), indicating strong intestinal absorption and drug-like properties. However, its moderate metabolic instability and predicted hepatotoxic potential suggest caution regarding systemic exposure and first-pass metabolism. Thus, the combination of computational and microbiological data reinforces the potential of CMA4DMA as an antimicrobial adjuvant to restore antibiotic efficacy against resistant strains, although additional studies on pharmacokinetics, toxicity, and preclinical models are required to confirm its therapeutic feasibility. In the future, we expect to carry out new in vivo and in vivo tests to corroborate the role of CMA4DMA as an inhibitor of the efflux pump mechanism, highlighting its role as an antibiotic adjuvant.

## Data Availability

Data is available from the authors upon reasonable request.

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
