# Peer review of "Evaluation of the Activity of Amino Chalcone Against Staphylococcus Strains Harboring Efflux Pumps"

_pharmaceuticals, 2025, doi:10.3390/ph18111629_

Round 1
Reviewer 1 Report
Comments and Suggestions for Authors
This article entitled “Evaluation of the activity of amino chalcone against Staphylococcus
strains harboring efflux pumps” is based on investigation of the synthetic chalcone CMA4DMA
as a potential modulator of bacterial resistance. In-vitro antimicrobial assays and fluorescence
assays were performed. In addition, the manuscript reports the in-silico analysis for the estimation
of binding energies of chalcones. Although the manuscript is attractive and well written still it
requires few major corrections that the authors need to incorporate before the manuscript is being
processed onward. Following are my comments about this manuscript.
Comments are:
1. The abstract seems less attractive. Most of the theoretical stuff have been given. For
making abstract attractive and clear in understanding, the results should be summarized in
quantitative term (numerical values). Authors are encouraged to rewrite the abstract.
2. The introduction section is providing limited information about threats caused by the
Staphylococcus aureus for which this whole research has been conducted. The authors are
encouraged to discuss some key threats caused by this microorganism and how this current
research is crucial.
3. In introduction section, authors are encouraged to provide a comprehensive mechanistic
explanation for the given sentence concept" acquisition of resistance genes via horizontal
gene transfer" that enhances bacterial resisting against any medication.
4. Enlist the natural as well as synthetic compounds that have been reported for exhibiting
potential in deactivation of efflux pumps. Also cite the relevant articles.
5. Page 3, Line 97: Provide full form of CCCP.
6. In Material and Method section, provide a scheme for synthesis of CMA4DMA.
7. Cite the referenced articles from where the method of substances preparation have adopted.
8. Page 16, Line 531 and 548: To provide good look to reader, the repetition of same statements
should be removed.
9. In Material and Method section, provide the equation for MIC calculation.
10. In Material and Method section, the paragraph on the general about instrumentation has not
been provided. It is important to provide the complete specifications of the instruments used
for this research.
11. In Conclusion section, future perspectives of the current research have not discussed. Authors
are encouraged to refine the conclusion section by providing key events of this whole research.
12. Please provide the structures of the investigated amino-chalcones.
13. In results section, MIC values and fluorescence % of investigated compounds and antibiotics
should be provided in tabulated form for better interpretation of research findings.
14. Page 16, line 533: Provided subheading should not be in the italic form.
15. In discussion section, from line 360 to 363, irrelevant data has been provided. The discussion
section is supposed to give in-depth explanation for the research findings. Please refine the
discussion section and only provide information about the current interpretation of your
results.
16. In discussion section, interpretation of observed fluorescence results from prospective of
previous studies have not provided. Authors are encouraged to compare the results with
previously reported data to make the discussion clear.
17. Docking results for investigated chalcones and the binding affinities have not compared well
with literature findings. Authors are encouraged to refine the discussion section.
Author Response
Reviewer 1
This article entitled “Evaluation of the activity of amino chalcone against Staphylococcus strains harboring efflux pumps” is based on investigation of the synthetic chalcone CMA4DMA as a potential modulator of bacterial resistance. In-vitro antimicrobial assays and fluorescence assays were performed. In addition, the manuscript reports the in-silico analysis for the estimation of binding energies of chalcones. Although the manuscript is attractive and well written still it requires few major corrections that the authors need to incorporate before the manuscript is being
processed onward. Following are my comments about this manuscript.
Dear reviewer, thank you for your consideration. I'm sure that after the adjustments the manuscript will be better.
Comments are:
1. The abstract seems less attractive. Most of the theoretical stuff have been given. For making abstract attractive and clear in understanding, the results should be summarized in quantitative term (numerical values). Authors are encouraged to rewrite the abstract.
R= We thank the reviewer for the valuable suggestion. In response, we have revised the abstract to make it more attractive and clear. Specifically, we have summarized the main results in quantitative terms, including key numerical values to provide a more precise and informative overview of our findings. We believe that the revised abstract now better highlights the significance and impact of the study.
2. The introduction section is providing limited information about threats caused by the
Staphylococcus aureus for which this whole research has been conducted. The authors are encouraged to discuss some key threats caused by this microorganism and how this current research is crucial.
R = Thank you for your valuable comments. All the suggested changes have been fully incorporated. The Introduction section was expanded to include a detailed discussion on the main threats posed by Staphylococcus aureus, emphasizing its high virulence, immune evasion capacity, and the broad spectrum of infections it can cause — ranging from mild skin lesions to severe conditions such as sepsis, pneumonia, and toxic shock syndrome.
3. In introduction section, authors are encouraged to provide a comprehensive mechanistic explanation for the given sentence concept" acquisition of resistance genes via horizontal gene transfer" that enhances bacterial resisting against any medication.
R= Thank you for your consideration. To strengthen the contextual understanding of resistance mechanisms, a comprehensive mechanistic explanation was added concerning the acquisition of resistance genes through horizontal gene transfer. This section now details the role of mobile genetic elements, including plasmids, transposons, pathogenicity islands, and staphylococcal chromosomal cassettes, in facilitating the rapid dissemination of resistance determinants
4. Enlist the natural as well as synthetic compounds that have been reported for exhibiting potential in deactivation of efflux pumps. Also cite the relevant articles.
R= Thank you for your consideration. An updated section listing natural and synthetic compounds described in the literature as potential efflux pump inhibitors (EPIs), along with their respective references, has been included. Synthetic compounds include PAβN, CCCP, NMP, and MBX2319; and natural compounds include silybin, boeravinone B, curcumin, berberine, columbamine, reserpine, and the essential oils of Origanum vulgare and Nigella
5. Page 3, Line 97: Provide full form of CCCP.
R= Thank you for your consideration, correction made.
6. In Material and Method section, provide a scheme for synthesis of CMA4DMA.
R= Correction made.
7. Cite the referenced articles from where the method of substances preparation have adopted.
R= Thank you for your consideration, correction made.
8. Page 16, Line 531 and 548: To provide good look to reader, the repetition of same statements should be removed.
R= Thank you for your considerations, correction made.
9. In Material and Method section, provide the equation for MIC calculation.
R=Dear reviewer, the MIC is performed only by calculating the geometric mean of the concentrations where inhibition of bacterial growth was achieved, with no further calculation being necessary.
10. In Material and Method section, the paragraph on the general about instrumentation has not been provided. It is important to provide the complete specifications of the instruments used for this research.
R= Dear reviewer, the equipment in each method was mentioned.
11. In Conclusion section, future perspectives of the current research have not discussed. Authors are encouraged to refine the conclusion section by providing key events of this whole research.
R= Dear reviewer, thank you for your considerations regarding the review.
12. Please provide the structures of the investigated amino-chalcones.
R= Correction made
13. In results section, MIC values and fluorescence % of investigated compounds and antibiotics should be provided in tabulated form for better interpretation of research findings.
R= Correction made
14. Page 16, line 533: Provided subheading should not be in the italic form.
R= Correction made
15. In discussion section, from line 360 to 363, irrelevant data has been provided. The discussion section is supposed to give in-depth explanation for the research findings. Please refine the discussion section and only provide information about the current interpretation of your results.
R=Dear referee, the mention of direct or indirect interaction with efflux pumps, of substances with efflux pumps, is related to the fact that interaction with membranes is a form of indirect inhibition of efflux pumps. In the case of CMA4DMA, this did not occur because it did not alter membrane permeability, which is why the aforementioned excerpt was cited.
16. In discussion section, interpretation of observed fluorescence results from prospective of previous studies have not provided. Authors are encouraged to compare the results with previously reported data to make the discussion clear.
R= Dear reviewer, thank you for your suggestion, correction made.
17. Docking results for investigated chalcones and the binding affinities have not compared well with literature findings. Authors are encouraged to refine the discussion section.
R= Dear referee, we appreciate your feedback regarding the molecular
docking simulations. We have considered your suggestion and made important
changes to the molecular docking discussion that we hope meet your most valuable review criteria.
Reviewer 2 Report
Comments and Suggestions for Authors
I have reviewed the manuscript and find it to be a well-structured, methodologically sound, and scientifically valuable contribution to the field of antimicrobial drug discovery. The manuscript is clearly written, the figures and tables are informative, and the conclusions are well supported by the data. The authors presented a compelling computational strategy in support of the experimental data. The following changes were suggested for the improvement of the manuscript.
1. MD simulations provide detailed insights on the binding mechanism and structural behaviour of the lead compound bound complex. Thus, authors may attempt to perform 300ns MD simulation, which will significantly enhance the findings.
2. Section 4.3. Molecular Docking of NorA and CMA4DMA. Which deals with the ADME and Drug-likeness analysis. Modify the subheadling.
3. From the protein structure used (PDB ID: 7LO7), it is evident that Quinolone resistance protein NorA is bound to a Fab25 Light Chain. However, it remains speculative whether the experimentally determined nature of this interaction with Fab25 Light Chain has been appropriately leveraged in the present in silico study. A critical point of clarification is whether the selected hit compounds bind to the same pocket observed in the X-ray structure or to a different site. This distinction needs to be explicitly addressed.
4. The docking procedure also requires further clarification. How was the docking protocol validated? Was any redocking performed to confirm accuracy and reproducibility? Furthermore, the absence of a control simulation is a notable limitation. Controls are essential to objectively compare the interaction quality of the hit compounds, enabling discrimination between strong, moderate, or weak binding profiles.
Author Response
I have reviewed the manuscript and find it to be a well-structured, methodologically sound, and scientifically valuable contribution to the field of antimicrobial drug discovery. The manuscript is clearly written, the figures and tables are informative, and the conclusions are well supported by the data. The authors presented a compelling computational strategy in support of the experimental data. The following changes were suggested for the improvement of the manuscript.
1. MD simulations provide detailed insights on the binding mechanism and structural behaviour of the lead compound bound complex. Thus, authors may attempt to perform 300ns MD simulation, which will significantly enhance the findings.
R=Dear referee, we are grateful for your suggestion. However, due to the
available computing resources and review time, we will not be able to perform
the MD simulations in a timely manner to meet your requirements. However, we
are considering a step that goes beyond the scope of this work, which seeks to
relate the static fitting pose to in vitro parameters, since the experiments
complement the theoretical data well. We would be extremely grateful if you
would consider the relevance of this work and the implementations we have
made to improve the exploration of the biological significance of the results in
your most prestigious review criteria.
2. Section 4.3. Molecular Docking of NorA and CMA4DMA. Which deals with the ADME and Drug-likeness analysis. Modify the subheadling.
R= Dear referee, we would like to thank you for your suggestion. We
have given the matter due consideration and implemented the requested
corrections.
3. From the protein structure used (PDB ID: 7LO7), it is evident that Quinolone resistance protein NorA is bound to a Fab25 Light Chain. However, it remains speculative whether the experimentally determined nature of this interaction with Fab25 Light Chain has been appropriately leveraged in the present in silico study. A critical point of clarification is whether the selected hit compounds bind to the same pocket observed in the X-ray structure or to a different site. This distinction needs to be explicitly addressed.
R= Dear referee, we appreciate your comment. In this new version, we
have compared the ligand-protein interactions that occur between the ligands
and the NorA efflux pump, in relation to Fab25, as requested.
4. The docking procedure also requires further clarification. How was the docking protocol validated? Was any redocking performed to confirm accuracy and reproducibility? Furthermore, the absence of a control simulation is a notable limitation. Controls are essential to objectively compare the interaction quality of the hit compounds, enabling discrimination between strong, moderate, or weak binding profiles.
R= Dear referee, in our study, we considered the fitting of chalcone at the
Fab25 site, as well as the fitting of Norfloxacin in relation to a previously
published molecular docking study using the same simulation protocol, as
validation of the simulation protocol. Due to the size of the ligand (Fab25) in
relation to conventional ligands used in simulations, it will not be possible to
replicate the same simulation protocol for Fab25. This is because it requires
more advanced computational resources. We would therefore be grateful if you
could give this new implementation due consideration when evaluating our
proposal.
Reviewer 3 Report
Comments and Suggestions for Authors
This research is highly innovative and has potential application value. The combination of computational and microbiological data reinforces the potential of CMA4DMA as an antimicrobial adjuvant to restore antibiotic efficacy against resistant strains such as Staphylococcus aureus.
Minor concerns:
- There is relatively little introduction to chalcone in the background. What makes CMA4DMA special and what is its current research status?
- P2L47, remove SYTOX in the Keywords part.
- P3L109, Figure 1, It is not recommended to break between 3 and 10.
- P4L120, Figure 2. The vertical axis is suggested to be extended to 75 units.
- P7, Is it possible to conduct some wet experimental research on ADMET?
Author Response
This research is highly innovative and has potential application value. The combination of computational and microbiological data reinforces the potential of CMA4DMA as an antimicrobial adjuvant to restore antibiotic efficacy against resistant strains such as Staphylococcus aureus.
Dear reviewer, thank you for your considerations. I'm sure that after the adjustments the manuscript will be better.
Minor concerns:
1. There is relatively little introduction to chalcone in the background. What makes CMA4DMA special and what is its current research status?
R= Correction made
2. P2L47, remove SYTOX in the Keywords part.
R= Correction made
3. P3L109, Figure 1, It is not recommended to break between 3 and 10.
R= Correction made
4. P4L120, Figure 2. The vertical axis is suggested to be extended to 75 units.
R= Correction made
5. P7, Is it possible to conduct some wet experimental research on ADMET?
R= Dear reviewer, we understand your concern, but the focus of our study is not the potential inhibitor of resistance mechanisms. In particular, we will be able to carry out a complete analysis of the in vivo toxicity of chalcones in another study.
Round 2
Reviewer 1 Report
Comments and Suggestions for Authors
The revised version of the manuscript article entitled “Evaluation of the activity of amino chalcone against Staphylococcus strains harboring efflux pumps" has been thoroughly reviewed for the required corrections. The authors have modified the manuscript according to the required suggestions. The abstract is concise and clearly stated now. The introduction encompasses all the relevant literature. The results and discussion are is comprehensive now. The references are relevant to the current research. The authors have responded positively to all the concern shown. The revised version is in much better form now. Therefore, I recommend this manuscript in this current form for publication in this journal.